# Generalized Animal Imitator:
# Agile Locomotion with Versatile Motion Prior

**Ruihan Yang**[1*], **Zhuoqun Chen**[1*], **Jianhan Ma**[1*], **Chongyi Zheng**[2*],
**Yiyu Chen**[3], **Quan Nguyen**[3], **Xiaolong Wang**[1]
[1]UC San Diego    [2]CMU    [3]USC

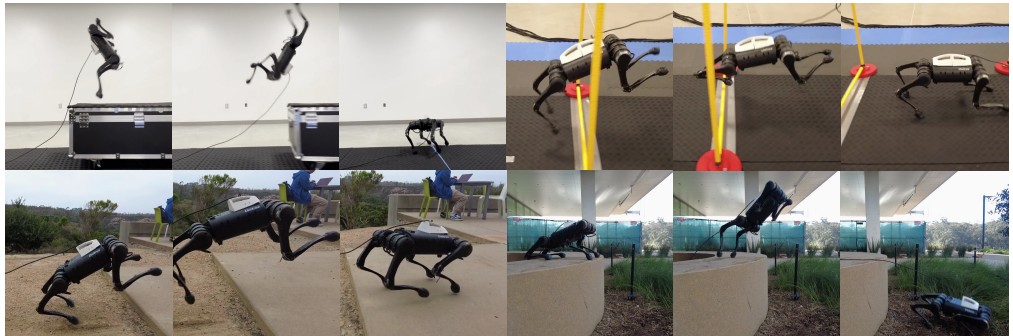

Figure 1: **Real-Robot Trajectory.** Our robot demonstrates diverse agile locomotion skills, including running, jumping, and back-flipping in real using a single motion prior and without fine-tuning.

**Abstract:** The agility of animals, particularly in complex activities such as running, turning, jumping, and backflipping, stands as an exemplar for robotic system design. Transferring this suite of behaviors to legged robotic systems introduces essential inquiries: How can a robot learn multiple locomotion behaviors simultaneously? How can the robot execute these tasks with a smooth transition? How to integrate these skills for wide-range applications? This paper introduces the Versatile Instructable Motion prior (*VIM*) – a Reinforcement Learning framework designed to incorporate a range of agile locomotion tasks suitable for advanced robotic applications. Our framework enables legged robots to learn diverse agile low-level skills by imitating animal motions and manually designed motions. Our *Functionality* reward guides the robot's ability to adopt varied skills, and our *Stylization* reward ensures that robot motions align with reference motions. Our evaluations of the VIM framework span both simulation and the real world. Our framework allows a robot to concurrently learn diverse agile locomotion skills using a single learning-based controller in the real world. Videos can be found on our website: https://generalizedanimalimitator.github.io

**Keywords:** Legged Robots, Imitation Learning, Agile Locomotion

## 1 Introduction

Researchers have been studying for years equipping legged robots with agility comparable to that of natural quadrupeds. Picture a golden retriever gracefully maneuvering in a park: darting, leaping over obstacles, and pursuing a thrown ball. These tasks, effortlessly performed by many animals, remain challenging for contemporary legged robots. To accomplish such tasks, robots need not only master individual agile locomotion skills like running and jumping, but also the capacity to adaptively select and configure these skills based on sensory inputs. The inherent ability of quadrupeds to smoothly execute diverse locomotion skills across varied tasks inspires our pursuit of a control system with

---

*Equal Contributions

8th Conference on Robot Learning (CoRL 2024), Munich, Germany.

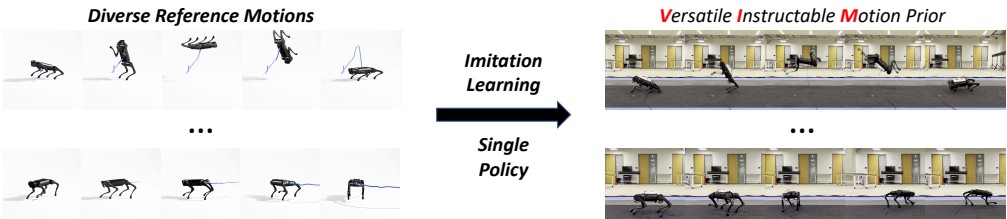

Figure 2: Our system learns a single instructable motion prior, from a diverse reference motion dataset.

Table 1: **Comparison of Skill Learning Framework. (More Details in Appendix D)**

| | Function Tracking | Style Tracking | Agility | Control Skills to Learn | Multiple Skills | Diverse Sources | Reusable | No Privileged Information | Real Deployment |
|---|---|---|---|---|---|---|---|---|---|
| Peng *et al.* [4] | ✓ | ✓ | ✗ | ✓ | ✗ | ✗ | ✗ | ✗ | ✓ |
| AMP [22] | ✗ | ✓ | ✗ | ✗ | ✓ | ✗ | ✗ | ✓ | ✓ |
| WASABI [23] | ✓ | ✗ | ✓ | ✓ | ✗ | ✗ | ✗ | ✗ | ✓ |
| ASE [18] | ✗ | ✓ | ✗ | ✗ | ✓ | ✗ | ✓ | ✓ | ✗ |
| Motion Imitation [17] | ✓ | ✓ | ✗ | ✓ | ✓ | ✗ | ✓ | ✓ | ✓ |
| **VIM** | ✓ | ✓ | ✓ | ✓ | ✓ | ✓ | ✓ | ✓ | ✓ |

a general locomotion motion prior that includes these skills. We introduce a novel RL framework, Versatile Instructable Motion prior (*VIM*) aiming to endow legged robots with a spectrum of reusable agile locomotion skills by integrating existing agile locomotion knowledge.

Agile gaits[1, 2, 3] for legged robots have been sculpted using model-based or optimization methods at the price of demanding significant engineering input and precise state estimation. Imitation-based controllers are also proposed to learn from motion sequences from animals [4] or optimization methods [5]. Recent works [6, 7, 8, 9, 10, 11, 12, 13, 14] also incoporate perception for legged robots. Despite encouraging results, most of these works focus on building a single controller from scratch, even though much of the learned locomotion skills could be shared across tasks. Recent works build reconfigurable low-level motion priors [15, 16, 17, 18, 19, 20] for downstream applications, but fail to make the best use of existing skills to learn diverse locomotion skills with high agility.

In this work, we focus on building low-level motion prior to utilize existing locomotion skills in nature and previous optimization methods, and learn multiple highly agile locomotion skills simultaneously, as in Figure 2. We utilize motion sequences to offer a consistent representation of diverse agile locomotion skills. Our motion prior extracts and assimilates a range of locomotion skills from reference motions, effectively mirroring their dynamics. These references comprise motion capture (mocap) sequences from quadrupeds, synthetic motion sequences complementing mocap data, and optimized motion trajectories. We translate varied reference motion clips into a unified latent command space, guiding the motion prior to recreate locomotion skills based on these latent commands and the robot's state. For legged robots, a locomotion skill is the ability to produce a specific trajectory. We classify this into two aspects: *Functionality* and *Style*. *Functionality* involves fundamental movement objectives, like moving forward at a set speed, while *Style* focuses on how a robot accomplishes a task, for example, two robots could run at the same speed, but with different gait. Teaching both aspects simultaneously is challenging [21]. We use three feedback: objective performance metrics, qualitative assessments, and detailed kinematic guidance. This structured approach helps the robot master basic functional objectives before refining its locomotion gaits.

By incorporating diverse reference motions and our reward design, our *VIM* learns diverse agile locomotion skills and makes them available for intricate downstream tasks. We evaluate our method in the simulation and real world, as Figure 1. Our method significantly outperforms baselines in terms of final performance and sample efficiency.

## 2   Related Work

**Blind Legged Locomotion:** Classical legged locomotion controllers [24, 25, 26, 27, 1, 28] based on model-based methods [29, 30, 31, 32, 33, 34, 35] and trajectory optimization [36, 3] have shown promising results in diverse tasks with high levels of agility. Nonetheless, these methods normally come with considerable engineering work for the specific task, high computation requirements during deployment, or fragility to complex dynamics. Learning-based methods [6, 37, 13, 38, 39, 40, 41]

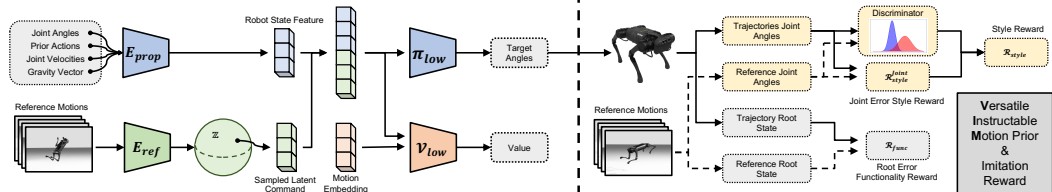

Figure 3: **VIM and Reward:** Our reference motion encoder maps reference motions into latent skill space and low-level policy output motor command. $V_{low}$ is the low-level critic for RL training. Our reward encourages the robot to track the root trajectory and the joint motion of the reference motion.

controllers are proposed to offer robust and lightweight controllers for deployment at the cost of offline computation. Peng et al [42] developed a controller producing non-agile life-like gaits by imitating animals. Though previous works offer robust or agile locomotion controllers across complex environments, these works focus on finishing a single task at a time without reusing previous experience. Peng et al [22] leverage reference motions as prior knowledge when directly addressing specific tasks. Li et al [23] obtain an agile locomotion skill from a single partial rough demonstration including only robot root trajectory. Smith et al [43] utilize existing locomotion skills to solve specific downstream tasks. Hoeller et al [44] built multiple individual locomotion skills and utilized them for agile navigation. Vollenweider et al [45] utilize multiple AMP [22] to develop a controller to solve a fixed task set. In this paper, our motion prior captures diverse agile locomotion skills from reference motions including mocap trajectories and trajectories generated by trajectory optimization, and provides them for intricate future downstream tasks.

**Motion Priors:** Due to the low sample efficiency and considerable effort required for reward engineering of RL, low-level skill pre-training has drawn growing attention. Singh et al [15] utilize a flow-based model to build an actionable motion prior with motion sequences generated by scripts. More recent works [16, 17, 18, 19, 46, 20, 47] focus on building low-level motion prior for downstream tasks but fail to include diverse highly agile locomotion skills. Luo et al [48, 49] developed unified motion prior for simulated humanoid robot. Peng et al [18] develop a simulation-based low-level motion prior entirely through unsupervised methods, yet they do not assure the acquisition of agile skills from the reference motion dataset (Additional discussion in Appendix C). In this work, we build motion prior with reference motions consisting of mocap sequences, synthesized motion sequences, and trajectories from optimization methods and learn multiple highly agile locomotion skills with a single controller. Comprehensive comparison is provided in Table 1.

# 3 Learn Versatile Instructable Motion Prior (*VIM*)

Building **V**ersatile **I**nstructable **M**otion prior (*VIM*), as shown in Figure 3, involves: constructing a reference motion dataset, and training the motion prior with an imitation-based reward system.

**Reference motion dataset:** Our dataset includes $N$ reference motions for various locomotion skills such as cantering, turning, backflips, and jumps. Reference motions are from: *(a)* Mocap data [50] of quadrupeds; *(b)* synthesized motions from a generative model[50] to enhance diversity; *(c)* motions from trajectory optimization methods. While mocap and synthesized motions provide extensive data, not all are feasible for the robot. Thus, trajectory-optimized motions are included for complex moves like jumps and backflips. The detailed motion list is in the Appendix A. To address differences between animals and our robot, we retarget mocap and synthesized sequences as per Peng et al. [4]. Each trajectory is noted as $(s_0^{\mathrm{ref}}, \cdots, s_T^{\mathrm{ref}})$, where $s_i^{\mathrm{ref}}$ is the reference robot state at $i$th timestep, and $T$ is the length of the reference motion. We denote the dataset as $\mathcal{D} = \{(s_0^{\mathrm{ref}}, \cdots, s_T^{\mathrm{ref}})_i\}_{i=1}^N$. Each frame includes the robot's pose, velocity, foot position, foot height, and joint angle and velocity without motor commands. Privileged information like robot pose and velocity is used only in simulation, and policies do not require it in the real world.

## 3.1 Motion Prior Structure

Our motion prior consists of a reference motion encoder and a low-level policy. Reference motion encoder maps varying reference motions into a condensed latent skill space, and low-level policy utilizes our imitation reward and reproduces the robot motion given a latent command.

**Reference motion encoder:** Our reference motion encoder $\mathbb{E}_{\text{ref}}(\cdot)$ maps segments of reference motion to latent commands in a latent skill space that outlines the robot's prospective movement. These segments are expressed as $\hat{s}_t^{\text{ref}} = \{s_{t+1}^{\text{ref}}, s_{t+2}^{\text{ref}}, s_{t+5}^{\text{ref}}, s_{t+10}^{\text{ref}}, s_{t+30}^{\text{ref}}\}$. Specifically, we choose $s_{t+1}^{\text{ref}}, s_{t+2}^{\text{ref}}, s_{t+5}^{\text{ref}}$ to provide immediate desired future motion, $s_{t+10}^{\text{ref}}, s_{t+30}^{\text{ref}}$ to provide desired motion over a longer time-span. We model the latent command as a Gaussian distribution $\mathcal{N}(\mathbb{E}_{\text{ref}}^{\mu}(\hat{s}_t^{\text{ref}}), \mathbb{E}_{\text{ref}}^{\sigma}(\hat{s}_t^{\text{ref}}))$ from which we draw a sample at each interval to guide the low-level policy. To maintain a temporal-consistent latent skill space, our training integrates an information bottleneck [51, 52] objective $L_{\text{AR}}$, where the prior follows an auto-regressive model [53]. Specifically, given the sampled latent command for the previous time step $z_{t-1}$, we minimize the KL divergence between the current latent Gaussian distribution and a Gaussian prior parameterized by $z_{t-1}$, $L_{\text{AR}}(\hat{s}_t^{\text{ref}}, z_{t-1}) = \beta \text{KL}\left(\mathcal{N}(\mu_t, \sigma_t^2) \parallel \mathcal{N}(\alpha z_{t-1}, (1-\alpha^2)I)\right)$ where $\alpha = 0.95$ is the scalar controlling the effect of correlation, $\beta$ is the coefficient balancing regularization.

**Low-level policy training:** Our low-level policy $\pi_{\text{low}}$ takes latent command $z_t$ and robot's proprioceptive state $s_t$ and outputs motor commands $a_t$ for the robot, where $s_t$ is encoded with a proprioception encoder $\mathbb{E}_{\text{prop}}$. We train low-level policy and reference motion encoder using PPO [54] in an end-to-end manner. We introduce learnable motion embeddings for the critic ($V_{low}$ in Figure 3) to distinguish reference motions. Episodes initiate with random frames of the dataset and terminate when the root pose tracking error is too large or the episode length is beyond the maximum length.

### 3.2 Imitation Reward for Functionality and Style

Given the formulation of our motion prior, the robot learns diverse agile locomotion skills with our imitation reward and reward scheduling mechanics. Our reward offers consistent guidance, ensuring the robot captures both the functionality and style inherent to the reference motion.

**Learning Skill Functionality:** To mirror the functionality of the reference motion, we translate the root pose discrepancy between agent trajectories and reference motion into a reward. The functionality reward $r_{\text{func}}$ includes tracking rewards for robot root position $r_{\text{func}}^{\text{pos}}$ and orientation $r_{\text{func}}^{\text{ori}}$. Recognizing the distinct importance of vertical movement in agile tasks, the root position tracking is further split into rewards for vertical $r_{\text{func}}^{\text{pos-z}}$ and horizontal movements $r_{\text{func}}^{\text{pos-xy}}$.

$$r_{\text{func}}(s_t, \hat{s}_t^{\text{ref}}) = w_{\text{func}}^{\text{ori}} * r_{\text{func}}^{\text{ori}} + w_{\text{func}}^{\text{pos-xy}} * r_{\text{func}}^{\text{pos-xy}} + w_{\text{func}}^{\text{pos-z}} * r_{\text{func}}^{\text{pos-z}}$$

The formulation of our functionality rewards is provided as follows, similar to previous work[4]:
$r_{\text{func}}^{\text{ori}}(s_t, \hat{s}_t^{\text{ref}}) = \exp\left(-10 \left\|\hat{\mathbf{q}}_t^{\text{root}} - \mathbf{q}_t^{\text{root}}\right\|^2\right)$, $r_{\text{func}}^{\text{pos-xy}}(s_t, \hat{s}_t^{\text{ref}}) = \exp\left(-20 \left\|\hat{\mathbf{x}}_t^{\text{root-xy}} - \mathbf{x}_t^{\text{root-xy}}\right\|^2\right)$,
$r_{\text{func}}^{\text{pos-z}}(s_t, \hat{s}_t^{\text{ref}}) = \exp\left(-80 \left\|\hat{\mathbf{x}}_t^{\text{root-z}} - \mathbf{x}_t^{\text{root-z}}\right\|^2\right)$ where $\mathbf{q}, \hat{\mathbf{q}}$ and $\mathbf{x}, \hat{\mathbf{x}}$ are the root orientation and position from the robot and reference motion respectively. Unlike previous work [4], we emphasize root height in our reward, crucial for mastering agile locomotion skills such as backflips and jumps.

**Learning Skill Style:** Capturing the style of a reference motion, in addition to its functionality, expands the application of the locomotion skills by meeting criteria such as energy efficiency, and robot safety. Drawing inspiration from how humans learn [55, 56] - starting by emulating the broader style before focusing on intricate joint movements - our robot first mimics the broader locomotion style with an adversarial style reward and later refines its technique with a joint angle tracking reward.

**Adversarial Stylization Reward:** We train discriminators $D_i$, $i = 1..N$ for all $N$ reference motions separately to distinguish robot transitions from the transition of reference motion [22, 45] and use the output to provide high-level feedback to the agent. Our discriminator is trained with:

$$\underset{D_i}{\text{argmin}} \quad \underset{d_i^{\mathcal{M}}(s_t, s_{t+1})}{\mathbb{E}} \left(D_i(s_t, s_{t+1}) - 1\right)^2 + \underset{d_i^{\pi}(s_t, s_{t+1})}{\mathbb{E}} \left(D_i(s_t, s_{t+1}) + 1\right)^2$$

where $d_i^{\mathcal{M}}(s_t, s_{t+1})$ and $d_i^{\pi}(s_t, s_{t+1})$ denote the state transition pair distribution of the dataset, and the state transition pair distribution generated by the policy for $i$th reference motion respectively. For each reference motion, the likelihood from the discriminator is then converted to a reward with: $r_{\text{style}}^{\text{adv}}(s_t, s_{t+1}) = 1 - \frac{1}{4} * (1 - D(s_t, s_{t+1}))^2$ Initially, our adversarial stylization reward provides dense reward and enables the robot to learn a credible gait, but it can not provide more detailed instructions as the training proceeds, which leads to mode collapse and unstable training.

Table 2: **Evaluation of Motion Prior in Simulation:** We compare Horizontal and Vertical Root Position (Root Pos (XY), Root Pos (Height)), Root Orientation (Root Ori), Joint Angle, and End Effector Position (EE Pos) tracking errors and RL objectives of all methods. Our methods outperform all baselines in terms of smaller tracking errors, higher episodic returns, and longer episode lengths. GAIL baseline shows a smaller root position tracking error since it can't follow the reference motion leading to early termination of the episode.

| Method | Tracking Error ↓ | | | | | RL Objectives ↑ | |
| | Root Pos (XY) (m$^2$) | Root Pos (Height) (m$^2$) | Root Ori (rad$^2$) | Joint Angle (rad$^2$) | EE Pos (m$^2$) | Episode Return | Episode Length |
| --- | --- | --- | --- | --- | --- | --- | --- |
| VIM | **1.24**±0.62 | 0.01±0.02 | 0.11±0.06 | **0.08**±0.06 | **0.03**±0.03 | 13.313±11.48 | 166.783±120.217 |
| VIM (w/o Scheduling) | 1.28±0.67 | 0.009±0.0123 | **0.1**±0.06 | 0.1±0.08 | 0.05±0.04 | **13.963**±11.395 | **179.047**±121.788 |
| Motion Imitation | 1.39±0.66 | **0.0077**±0.0114 | 0.11±0.05 | 0.25±0.14 | 0.14±0.08 | 9.536±9.049 | 143.393±114.514 |
| GAIL | 1.04±0.86 | 0.03±0.03 | 0.13±0.05 | 0.17±0.1 | 0.09±0.05 | 3.586±6.166 | 54.723±75.984 |
| WASABI | 0.54±0.68 | 0.03±0.03 | 0.13±0.06 | 4.14±1.06 | 0.21±0.07 | 0.71±0.58 | 22.82±16.13 |
| VIM (w/o Func Reward) | 1.24±0.67 | 0.01±0.02 | 0.11±0.06 | 0.61±0.59 | 0.02±0.02 | 10.66±12.92 | 115.19±117.74 |
| VIM (w/o Style Reward) | 1.49±0.69 | 0.00±0.01 | 0.12±0.06 | 5.14±1.71 | 0.25±0.06 | 6.28±6.73 | 109.76±110.67 |

**Joint Angle Tracking Reward:** On the contrary, joint angle tracking reward [57, 17] provides stable instruction for the robot to mimic the gait of reference motion, while the reward signal is small in scale during the initial training stage when the joint angle is away from joint target. Similar to our root pose tracking reward, our joint angle tracking reward has the following formulation:

$$r_{style}^{joint}(s_t, \hat{s}_t^{ref}) = \exp\left(-5\sum_{j\in joints}\left\|\hat{q}_t^j - q_t^j\right\|^2\right) + \exp\left(-20\sum_{f\in feet}\left\|\hat{e}_t^f - e_t^f\right\|^2\right) + \exp\left(-20\sum_{f\in feet}\left\|\hat{h}_t^f - h_t^f\right\|^2\right)$$

where $q_t^j, \hat{q}_t^j$ are the joint angle of robot and reference motion, $e_t^f, \hat{e}_t^f$ are the end-effector positions of robot and reference motion, $h_t^f, \hat{h}_t^f$ are the end-effector height of robot and reference motion.

**Stylization Reward Scheduling:** To learn the style quickly and stably, we propose to use both adversarial stylization reward and joint angle tracking reward with a balanced scheduling mechanism. Considering the discriminator as a "coach", We utilize the mean adversarial reward as an indication of how the coach is satisfied with the current performance. When it's not satisfied with the current performance of the robot, it provides detailed instructions for the robot to learn. Specifically, our stylization reward follows: $r_{style}(s_t, \hat{s}_t^{ref}) = w_{style}^{adv} * r_{style}^{adv} + w_{style}^{joint} * r_{style}^{joint} + w_{style}^{adv} * (1 - \mathbb{E}_{s_t \in S}(r_{style}^{adv}(s_t, s_{t+1}))) * r_{style}^{joint}$

With this formulation, our stylization reward provides dense rewards at the beginning of training, enabling the robot to quickly catch the essence of different agile locomotion skills, and provides detailed instruction as the training proceeds, enabling the robot to refine its gait.

### 3.3 Solving Downstream Tasks with Motion Prior:

For intricate tasks like jumping over gaps, starting from scratch is challenging due to the need for agile locomotion skills and the intensive engineering to balance rewards and regularize motion. Using a low-level motion prior, robots can immediately use existing skills and focus on high-level strategies. For each distinct downstream task, we train a high-level policy $\pi_{high}$ (As shown in Figure 4) takes the high-level observation $o_{high}$, and proprioception of the robot and outputs latent command for low-level motion prior to utilize the existing skills: $a_t = \pi_{low}(\pi_{high(o_{high}, s_t)}, E_{prop}(s_t))$.

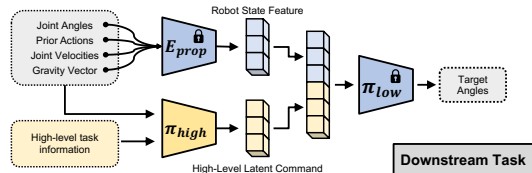

Figure 4: **Solving High-level Tasks with Our Motion Prior.** Our high-level policy outputs high-level latent command for low-level policy.

Additional implementation details about observation/action space, reference/proprioception encoder, low-level/high-level policy, and value network can be found in the Appendix F.

## 4 Experiments

We evaluate our system in simulation and real-world, comparing with prior work for low-level skill learning and various high-level tasks. Our robot demonstrates life-like agility in the real world.

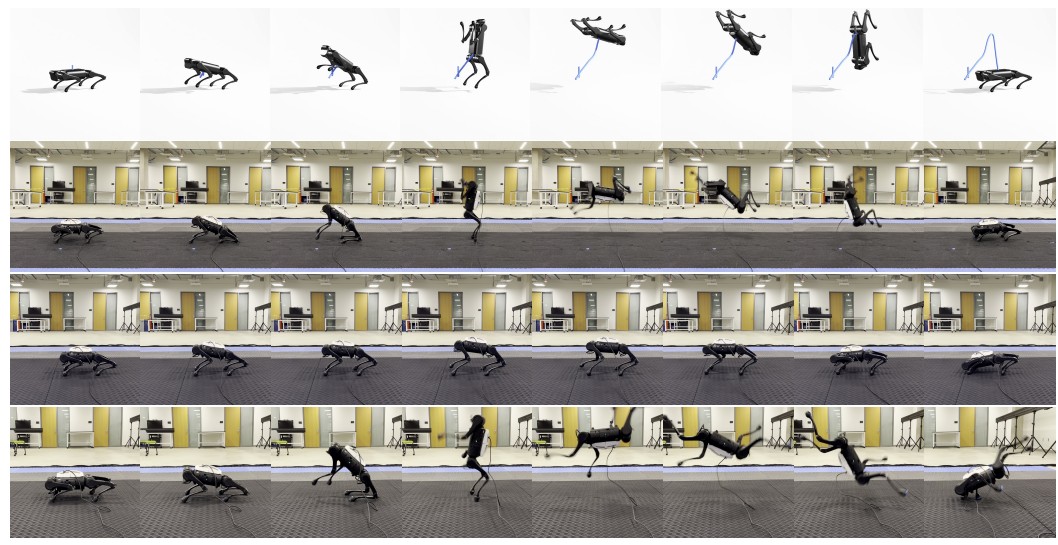

Figure 5: **Real World `Backflip` Trajectory:** Each row represents a single trajectory (From top to bottom: Reference Motion, VIM, GAIL, Motion Imitation). Trajectories are shown from left to right.

Table 3: **Evaluation of Motion Prior in Real:** We collect representative metrics for different skills with corresponding metrics from reference motion. $N/A$ denotes completely failed skills in real.

| Metrics | Unit | VIM | Motion Imitation | GAIL | Reference Motion |
|---|---|---|---|---|---|
| Height (Jump While Running) | $(m)$ | $\mathbf{0.50}_{\pm 0.003}$ | $0.42_{\pm 0.01}$ | $0.41_{\pm 0.04}$ | $0.53_{\pm 0.005}$ |
| Height (Jump Forward) | $(m)$ | $\mathbf{0.44}_{\pm 0.01}$ | $0.42_{\pm 0.01}$ | $0.27_{\pm 0.006}$ | $0.59_{\pm 0.006}$ |
| Height (Jump Forward (Syn)) | $(m)$ | $\mathbf{0.52}_{\pm 0.01}$ | $N/A$ | $N/A$ | $0.55_{\pm 0.007}$ |
| Height (Backflip) | $(m)$ | $\mathbf{0.62}_{\pm 0.01}$ | $0.49_{\pm 0.01}$ | $N/A$ | $0.60_{\pm 0.005}$ |
| Distance (Jump While Running) | $(m)$ | $\mathbf{0.48}_{\pm 0.08}$ | $0.35_{\pm 0.02}$ | $0.40_{\pm 0.003}$ | $0.56_{\pm 0.008}$ |
| Distance (Jump Forward) | $(m)$ | $\mathbf{0.76}_{\pm 0.05}$ | $0.40_{\pm 0.01}$ | $0.10_{\pm 0.002}$ | $0.82_{\pm 0.003}$ |
| Distance (Jump Forward (Syn)) | $(m)$ | $\mathbf{0.49}_{\pm 0.04}$ | $N/A$ | $N/A$ | $0.54_{\pm 0.007}$ |
| Linear Velocity (Pace) | $(m/s)$ | $\mathbf{0.76}_{\pm 0.01}$ | $0.97_{\pm 0.07}$ | $0.50_{\pm 0.02}$ | $0.72_{\pm 0.05}$ |
| Linear Velocity (Canter) | $(m/s)$ | $\mathbf{1.49}_{\pm 0.15}$ | $N/A$ | $N/A$ | $3.87_{\pm 0.17}$ |
| Linear Velocity (Walk) | $(m/s)$ | $0.90_{\pm 0.04}$ | $\mathbf{0.96}_{\pm 0.06}$ | $0.53_{\pm 0.58}$ | $0.97_{\pm 0.42}$ |
| Linear Velocity (Trot) | $(m/s)$ | $1.33_{\pm 0.17}$ | $\mathbf{1.05}_{\pm 0.02}$ | $0.93_{\pm 0.01}$ | $1.16_{\pm 0.12}$ |
| Angular Velocity (Left Turn) | $(rad/s)$ | $1.71_{\pm 0.04}$ | $0.00_{\pm 0.00}$ | $\mathbf{0.91}_{\pm 0.04}$ | $1.01_{\pm 0.05}$ |
| Angular Velocity (Right Turn) | $(rad/s)$ | $0.81_{\pm 0.02}$ | $\mathbf{0.62}_{\pm 0.02}$ | $0.63_{\pm 0.05}$ | $0.41_{\pm 0.09}$ |
| Joint Angle Tracking Error | $(rad^2/joint)$ | $\mathbf{0.10}_{\pm 0.08}$ | $0.27_{\pm 0.16}$ | $0.22_{\pm 0.10}$ | - |

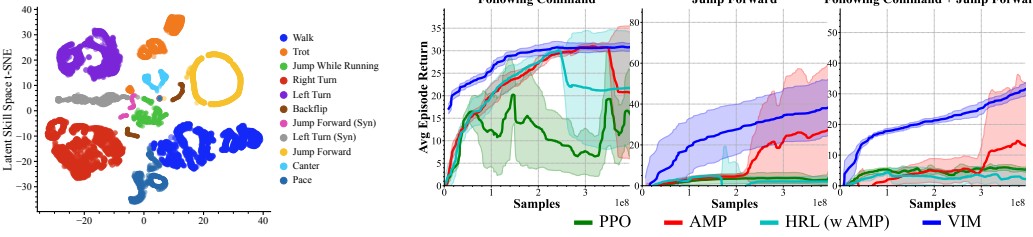

Figure 6: **Latent Skill Space t-SNE.** We visualize the latent embedding for varying motion segments.

Figure 7: **High-level Tasks Evaluation in Simulation:** Solid line and shaded area denote the mean and std across random seeds. Our system outperforms all baselines.

## 4.1 Evaluation of Learned Low-level Motion Priors

**Baselines**: We benchmark our method against three representative baselines: **Motion Imitation** [4, 17, 20] baseline represents a thread of recent works whose imitation rewards are defined solely with errors between current robot states and the corresponding reference states. Generative Adversarial Imitation Learning (**GAIL**) baseline represents a thread of recent work [18], whose imitation reward is solely provided by the discriminator trained to distinguish trajectories generated by the policy from the ground truth reference motions. **WASABI** baseline represents a modified version of WASABI [23] for our setting. Each method trains for $2 \times 10^9$ samples across 3 random seeds. Both our method and the Motion Imitation baseline adopt identical reward scales for all motion error-tracking rewards.

Table 4: **High-level Tasks in Real World:** We compare `Following Command + Jump Forward` policies of all methods in real. $N/A$ denotes completely failed skills in real. Our methods outperform all baselines in real for most metrics.

| Metrics | Unit | Ours | AMP | PPO | HRL |
|---|---|---|---|---|---|
| Max Linear Velocity | $(m/s)$ | **1.78**±**0.13** | 1.74±0.21 | 1.75±0.26 | 1.70±0.08 |
| Max Angular Velocity (Left) | $(rad/s)$ | 1.78±0.004 | 1.07±0.09 | **2.24**±**0.05** | 0.00±0.00 |
| Max Angular Velocity (Right) | $(rad/s)$ | **2.05**±**0.02** | 0.83±0.09 | 1.75±0.19 | 0.95±0.37 |
| Jump Distance | $(m)$ | **0.50**±**0.07** | 0.00±0.00 | $N/A$ | $N/A$ |
| Jump Height | $(m)$ | **0.50**±**0.02** | 0.38±0.01 | $N/A$ | $N/A$ |

**Simulation Evaluation:** In the simulation, we measure average imitation tracking errors, episode returns, and trajectory lengths across random seeds. As listed in Table 2, the tracking error of root pose represents the ability of the robot to reproduce the locomotion skill, and the tracking error of joint angle and end effector position represents the ability of the robot to mimic the style of reference motion. Our method achieves a similar root pose tracking error as the motion imitation baseline with a much smaller joint angle tracking error. This shows that our method strikes a balance between functionality and style, superior to the motion imitation baseline that focuses mainly on functionality. Meanwhile, the GAIL baseline failed to learn the functionality of the reference motions leading to short episode length and the least episode return. We surmise that the GAIL baseline's inadequacy arises from the adversarial reward does not offer temporally consistent guidance throughout skill learning and the mode collapse issue inherent in adversarial training hinders the robot from mastering highly agile skills, such as backflipping. The poor performance of the Motion Imitation baseline may stem from the challenges of balancing different terms and selecting suitable hyperparameters when concurrently learning multiple agile locomotion skills.

**Ablation Study of Learned Motion Prior:** We provide the ablation study over the reward term and the scheduling mechanism as shown in Table 2. We found that without *Functionality* reward, the learned controller could not robustly track the reference motion resulting in smaller Episode Return and, shorter Episode Length on the other hand, removing *Style* reward results in a significantly higher Joint Angle and End-Effector tracking error. Comparing VIM with and without stylization reward scheduling, we find the former exhibits enhanced style tracking performance, underscoring the value of stylization reward scheduling in refining robot gait tracking.

**Real World Evaluation:** We evaluated learned agile locomotion skills in the real world using specific metrics tailored to different skills, as detailed in Table 3. We repeated our experiment three times per skill per method per seed since our real-world experiment. For `Jump While Running/Jump Forward/Jump Forward (Syn)/Backflip`, we measured jumping height and distance. For `Pace/Canter/Walk/Trot` and `Left Turn/Right Turn`, we measured linear and angular velocity. Results show our method retains most of the reference motion functionality. The only significant deviation observed in `Canter` is due to differences between animal and robot capabilities, as quadrupeds use tendons to achieve higher running speeds, which our robot lacks. Despite similar root pose tracking errors in simulations, our method outperforms the Motion Imitation baseline in real-world metrics like jumping height, distance, and velocity tracking error, indicating that mirroring reference motion style improves sim2real transfer for natural gaits. The GAIL baseline struggled with real-world locomotion skills. Figure 5 visually compares real-world trajectories, showing our method's superiority in capturing both motion functionality and style. Due to poor simulation performance, the WASABI baseline was not evaluated in the real world.

**Latent Skill Space Visualization:** We visualize the learned latent skill space in Figure 6 by visualizing the latent embedding corresponding to motion segments in our reference motion dataset via t-SNE [58]. We find that different skills are separated into different regions with clear boundaries. Our reference motion encoder also clusters the skills with similar semantic meaning together: embeddings from `Left Turn`/`Right Turn` sequence are close, which enables the smooth transition between different skills. embeddings from `Jump While Running` & `Jump Forward` & `Jump Forward (Syn)` sequence are clustered together. These observations suggest that our system learned a smooth and semantically meaningful latent skill space for solving high-level tasks.

### 4.2 Evaluation on High-level Tasks

To evaluate how our method leverages learned agile locomotion skills for high-level tasks, we designed a set of tasks and tested our method against baselines in simulation and the real world.

**High-level Tasks & Observation:** Our tasks include: `Following Command`: directing the robot to move with specific linear and angular velocities. Linear velocity commands range from $0 \sim 2$ m/s, and angular velocity commands range from $-2 \sim 2$ rads/s. In our motion prior, the robot is trained to move and turn at the reference motion's speed. Hence, to follow a command precisely, the high-level policy needs to smoothly interpolate between different speeds. `Jump Forward`: directing the robot to jump while running. We have adapted a subset of jumping rewards from CAJun [59] to evaluate policy interpolation between jumping and running motions within a fixed timeframe. `Following Command + Jump Forward`: directing the robot to either jump forward or adjust to changing commanded speeds. To optimize episode return, the robot should not only use the agile locomotion skills from the reference motion dataset but also develop unobserved skills like executing sharp turns. Detailed high-level observations for different tasks are provided in the Appendix F.

**Baselines:** Given the baseline's subpar performance in low-level motion prior training, we compare our system with three representative baselines without pre-trained low-level controllers: **PPO** [54]: Controllers trained exclusively on high-level task rewards. **AMP** [22]: Utilizes reference motion for styling reward in adversarial imitation learning and learns high-level tasks while mimicking reference motions. **Hierarchical Reinforcement Learning (HRL)** from Jain et al. [60]: Learns a high-level policy sending latent commands to a low-level controller, resembling works that decompose tasks into sub-problems [61, 62, 63, 64, 65, 66, 67]. For fair comparison, we removed the trajectory generator in [60], used PPO for AMP and HRL, and used full reference motion for AMP and HRL with AMP.

**Evaluation in Simulation & Real World:** We trained all methods on each high-level task for $4 \times 10^8$ samples with 3 random seeds. Simulation results are detailed in Figure 7, and real-world results are provided in Table 4. Real-world Following Commands trajectory is also provided in Appendix I. For the `Following Command` task, all methods mastered basic locomotion, but ours excelled in efficiency and smooth transitions between diverse linear and angular velocities. In the `Jump Forward` and `Following Command + Jump Forward` tasks, which required advanced jumping abilities, baselines struggled. They either moved forward continuously, remained grounded when prompted to jump, or toppled to avoid energy consumption penalties. In contrast, our system seamlessly integrated jumping and running actions, achieving the highest episode return. Despite having a comprehensive reference motion dataset, baselines couldn't harness the skills effectively. This likely stems from the difficulty of deriving agile locomotion skills using only adversarial stylization rewards, similar to the GAIL baseline's poor performance in low-level training.

## 5   Limitations

Our current system exhibits several limitations: 1) Safety is not assured with our current method. Introducing safety constraints during training could mitigate this issue; 2) The robot's limited capacity restricts its ability to fully replicate certain motion capture data, like cantering. Upgrading the hardware could address this limitation; 3) Currently, our system does not incorporate dynamics information. This could be improved by integrating adaptation techniques during deployment; 4) The system's low-level motion priors and high-level policies currently lack perceptual information.

## 6   Conclusion

In this paper, we propose Versatile Instructable Motion prior (*VIM*) which learns agile locomotion skills from diverse reference motions with a single motion prior. Our simulation and real-world results show that our VIM captures both the functionality and style of locomotion skills from reference motions. Our VIM also provides a temporally consistent and compact latent skill space representing different locomotion skills for high-level tasks. With agile locomotion skills in our VIM, complex High-level tasks can be solved efficiently with minimum human effort.

## Acknowledgement

This work was supported, in part, by NSF CCF-2112665 (TILOS), NSF 1730158 CI-New: Cognitive Hardware and Software Ecosystem Community Infrastructure (CHASE-CI), NSF ACI-1541349 CC*DNI Pacific Research Platform.

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

# Appendix

## A    Reference Motion Dataset

Our reference motions (11 reference motions in total) come from motion capture of animal motion, trajectory optimization method, and synthesized data with a generative model. The length of our reference motion ranges from 32 to 500. During training, we repeat the reference motions cyclically to fit the length of the episode.

Table 5: **Reference Motion Dataset:**

| Skill Name | Walk (Mocap) | Trot (Mocap) | Jump while Running (Mocap) | Right Turn (Mocap) | Left Turn (Mocap) | Pace (Mocap) |
|---|---|---|---|---|---|---|
| Motion length | 500 | 32 | 500 | 38 | 45 | 38 |

| Skill Name | Jump Forward (Synthetic) | Left Turn (Synthetic) | Backflip (Optimization) | Jump Forward (Optimization) | Canter (Mocap) |
|---|---|---|---|---|---|
| Motion length | 500 | 500 | 129 | 120 | 64 |

## B    Performance Across Different Reference Motions

In our framework, when the root pose of the simulated robot diverges too much from the reference root pose, we terminate the episode, as described in Sec 3.1 . In this case, the episode length is a good indicator of whether the learned policy could follow the reference motions. As shown in Fig. 8 , the performance of our framework varies when imitating different reference motions. When imitating relatively steady motions like *Walk (Mocap)*, *Pace (Mocap)*, *Left Turn (Mocap)*, the learned controller could track the motion for a longer period. When imitating relatively agile motions, especially with high moving speed, such as *Canter (Mocap)*, *Jump While Running (Mocap)*, the performance of our system drops. This phenomenon is rooted in the methodology disparities between our robot and real animals.

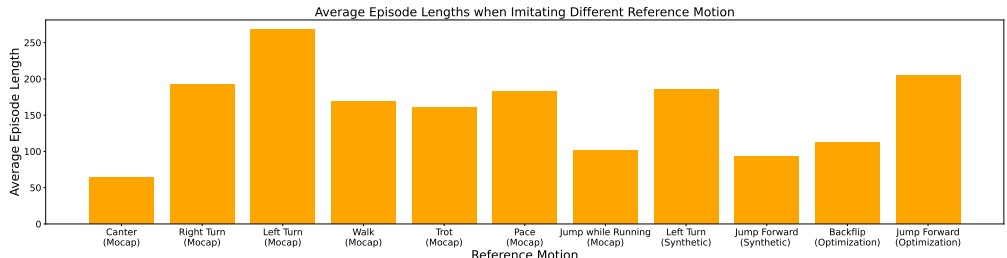

Figure 8: **Performance for different reference motions:** We provide the average episode length of the learned motion prior when it imitates different reference motions.

## C    Additional Discussion about ASE

We would like to clarify the significant differences between our method and the ASE baseline in the following aspects:

- **Skill Control and Learning:** Unlike ASE, which learns motor skills from a transition dataset in an unsupervised manner without controlling the outcome, our method intentionally learns specific motor skills from our reference motion dataset. This controlled learning approach ensures that critical skills, such as the jump motion are effectively acquired. This capability is crucial for constructing a motion prior tailored for varied high-level tasks. As shown in ASE video and demonstration even though there is jump motion in their dataset, ASE failed to learn it.

- **Long-Term Skill Acquisition:** ASE focuses on learning motor skills at the transition level, limiting its ability to learn complex, long-term motor skills like backflipping, which require extended motion sequences. Our method, however, leverages a combination of adversarial

styling and dense tracking rewards, providing structured supervision for acquiring long-term motor skills, including agile locomotion abilities like backflipping and jumping forward. This sequence-level modeling is essential for the effective learning of complex locomotion skills.

- **Performance Evaluation:** Since ASE learns different motor skills in an unsupervised manner, it's difficult to evaluate the performance of the learned low-level controller (There is no evaluation or benchmark for the low-level controller in ASE paper). While our method learns to imitate different locomotion skills in the dataset at the sequence level, we could directly benchmark the tracking error to evaluate the quality of the learned low-level controller. Benchmark over the low-level controller is also important if we want to build a backbone of low-level skills for diverse potential high-level tasks.

## D  Additional Discussion over Skill Learning Frameworks

In this section, we provide further discussion on the existing skill-learning framework:

- **Function Tracking:** The resulting controller of a skill-learning framework can accurately track the movement of the robot's base.
- **Skill Tracking:** The resulting controller can faithfully replicate the joint movement patterns of the robot.
- **Agility:** The controller is capable of producing highly agile locomotion skills. Since there is no universally accepted definition of "agile", in our work, we consider a skill "agile" if it involves the robot leaving the ground, such as in a backflip or jump, or running/turning at high speed.
- **Control Skills to Learn:** Given a fixed set of reference motions, the resulting controller can reliably reproduce specific skills. Unsupervised methods like ASE do not guarantee performance on any particular skill in the dataset.
- **Multiple Skills:** The resulting controller is capable of performing a variety of different skills.
- **Diverse Sources:** The resulting controller learns different skills from various sources.
- **Reusable:** The resulting controller can be reused for tasks beyond reproducing the reference motion's skills.
- **No Privileged Information:** The resulting controller does not require privileged information (such as the robot's velocity or position in the world frame) during deployment.
- **Real Deployment:** The proposed framework is validated in real-world scenarios.

**Additional discussion on the performance of ASE/AMP-based methods in agile locomotion skills:** ASE struggles to capture agile locomotion skills for the following reasons:

- **Difficulty in Learning Agile Skills:** Agile locomotion skills, such as jumping and backflipping, are inherently more challenging to learn compared to other locomotion skills like walking or trotting. Due to the well-known mode-collapse issue in the generative adversarial learning paradigm, it is particularly difficult for generative adversarial methods (like ASE/AMP) to discover and learn these complex skills in an unsupervised manner.
- **Limitations of Transition-Level Learning:** As discussed in Appendix C, ASE/AMP performs adversarial learning at the transition level, focusing on the current and previous states of the robot (as shown in Formula (3) in ASE[18]). However, skills that require a longer sequence of actions, such as backflipping or jumping, are difficult to learn with this approach. For example, executing a backflip involves multiple stages: sitting down, lifting the front legs, pushing off with the rear legs, adjusting the pose in mid-air, and landing. Similarly, jumping and running require coordinated stages of movement. Transition-level supervision lacks the long-term guidance needed to learn these complex, agile skills.

Table 6: **Evaluation of Motion Prior in Simulation for single Reference motion:** We compare Horizontal and Vertical Root Position (Root Pos (XY), Root Pos (Height)), Root Orientation (Root Ori), Joint Angle, and End Effector Position (EE Pos) tracking errors and RL objectives of all methods.

| Method | Tracking Error ↓ | | | | | RL Objectives ↑ | |
| --- | --- | --- | --- | --- | --- | --- | --- |
| | Root Pos (XY) $(m^2)$ | Root Pos (Height) $(m^2)$ | Root Ori $(rad^2)$ | Joint Angle $(rad^2)$ | EE Pos $(m^2)$ | Episode Return | Episode Length |
| *Jump Forward (Optimization)* | | | | | | | |
| VIM | $1.16_{\pm0.52}$ | $0.01_{\pm0.01}$ | $0.10_{\pm0.05}$ | $0.99_{\pm0.30}$ | $0.04_{\pm0.01}$ | $38.47_{\pm9.18}$ | $422.49_{\pm96.00}$ |
| Motion Imitation | $1.19_{\pm0.45}$ | $0.00_{\pm0.00}$ | $0.13_{\pm0.06}$ | $4.24_{\pm1.30}$ | $0.12_{\pm0.02}$ | $26.11_{\pm8.76}$ | $325.08_{\pm105.22}$ |
| GAIL (Single Skill AMP) | $2.00_{\pm0.48}$ | $0.04_{\pm0.01}$ | $0.10_{\pm0.05}$ | $0.92_{\pm0.22}$ | $0.03_{\pm0.01}$ | $11.71_{\pm6.59}$ | $161.38_{\pm83.92}$ |
| *Jump While Running (Mocap)* | | | | | | | |
| VIM | $1.58_{\pm0.56}$ | $0.01_{\pm0.01}$ | $0.09_{\pm0.03}$ | $1.63_{\pm0.18}$ | $0.06_{\pm0.01}$ | $13.36_{\pm7.16}$ | $172.90_{\pm90.81}$ |
| Motion Imitation | $1.50_{\pm0.49}$ | $0.00_{\pm0.00}$ | $0.09_{\pm0.03}$ | $3.04_{\pm0.99}$ | $0.13_{\pm0.06}$ | $10.93_{\pm5.08}$ | $163.36_{\pm74.99}$ |
| GAIL (Single Skill AMP) | $2.19_{\pm0.84}$ | $0.04_{\pm0.01}$ | $0.17_{\pm0.04}$ | $2.48_{\pm0.61}$ | $0.11_{\pm0.02}$ | $4.61_{\pm2.84}$ | $120.48_{\pm81.14}$ |
| *Trot (Mocap)* | | | | | | | |
| VIM | $1.21_{\pm0.32}$ | $0.00_{\pm0.00}$ | $0.08_{\pm0.04}$ | $0.18_{\pm0.03}$ | $0.01_{\pm0.00}$ | $21.52_{\pm10.58}$ | $213.10_{\pm103.26}$ |
| Motion Imitation | $1.21_{\pm0.29}$ | $0.00_{\pm0.00}$ | $0.10_{\pm0.05}$ | $0.17_{\pm0.03}$ | $0.01_{\pm0.00}$ | $17.62_{\pm8.86}$ | $174.88_{\pm87.40}$ |
| GAIL (Single Skill AMP) | $1.76_{\pm0.78}$ | $0.00_{\pm0.00}$ | $0.08_{\pm0.05}$ | $0.61_{\pm0.60}$ | $0.03_{\pm0.03}$ | $14.87_{\pm8.76}$ | $159.49_{\pm90.19}$ |
| *Left Turn (Mocap)* | | | | | | | |
| VIM | $0.07_{\pm0.08}$ | $0.00_{\pm0.00}$ | $0.16_{\pm0.07}$ | $0.15_{\pm0.02}$ | $0.01_{\pm0.00}$ | $31.64_{\pm14.21}$ | $299.63_{\pm135.47}$ |
| Motion Imitation | $0.11_{\pm0.12}$ | $0.00_{\pm0.00}$ | $0.14_{\pm0.07}$ | $0.60_{\pm0.41}$ | $0.03_{\pm0.02}$ | $35.31_{\pm14.05}$ | $383.56_{\pm135.10}$ |
| GAIL (Single Skill AMP) | $0.15_{\pm0.16}$ | $0.00_{\pm0.00}$ | $0.17_{\pm0.08}$ | $0.18_{\pm0.10}$ | $0.01_{\pm0.01}$ | $27.75_{\pm16.92}$ | $268.36_{\pm162.53}$ |

- **Limited Input for Discriminator:** ASE/AMP discriminators only consider joint angles as input, making it more challenging for the robot to learn agile locomotion skills that involve significant changes in the robot's position and orientation.

# E    Single Skill Comparison in Simulation

We conducted additional evaluations focusing on single skill learning, where all methods are required to learn a single reference motion using identical hyperparameters. For these experiments, we removed the motion embedding from the critic since there is only one skill to learn. We selected *Jump Forward (Optimization)*, and *Jump while Running (Mocap)* as representative skills for agile locomotion, and *Trot (Mocap)* and *Left Turn (Mocap)* as representative skills for normal locomotion. Each method was trained with $2 \times 10^9$ samples per skill across three random seeds.

In general, as shown in Table 6, single-skill tracking tends to deliver better results, in terms of longer episode length and higher episode return (representing the overall performance), for the specified skill because the task is easier to learn and more samples are dedicated to that particular skill. (In our low-level motion prior training stage, all skills share the total number of samples.) For agile locomotion skills, our method outperforms both GAIL (Single Skill AMP) in terms of better tracking of the robot's root movement and the motion imitation baseline by achieving smaller joint tracking errors. For normal locomotion skills, all methods deliver reasonable results for both *Trot (Mocap)* and *Left Turn (Mocap)*. However, GAIL (Single Skill AMP) exhibits slightly higher joint tracking error for Trot, which we attribute to the lack of temporal alignment in the adversarial reward. Although the GAIL baseline can faithfully reproduce the skill, it shows a slightly higher tracking error. It's important to note that the results in Table 6 should not be directly compared with those in Table 2 as longer episodes tend to accumulate more errors, leading to larger tracking errors, and the experiment setting is not identical.

# F    Implementation Details

**Observation & Action Space:** Our low-level observation includes joint angles, joint velocities, gravity vector in the robot frame, and the previously executed actions. Our controller outputs target joint angles for 12 joints of our robot in 25hz. The target joint angles are converted to torque command with PD controller where KP=40, KD=1.0.

**High-level Observation** For `Following Command` task, our high-level observation includes the target linear velocity and target angular velocity. For `Jump Forward` task, our high-level

observation includes the target jumping forward velocity and the normalized phase information in the jumping forward cycle. For `Following Command + Jump Forward` task, our high-level observation includes the high-level observation for both *Following Command* and *Jump Forward* tasks as well as an additional binary command indicating whether following command or jump forward at current time-step

**Reference Encoder $E_{ref}$ & Proprioception Encoder $E_{prop}$:** Our reference encoder proprioception encoder are both two-layer MLP with [256] hidden units, mapping the reference motion segment into a 64 dimensional latent distribution and proprioception into a 64-dim robot state feature, respectively.

**Low-level Policy $\pi_{low}$ and Value Network $V_{low}$:** Our low-level policy is a three-layer MLP with [256, 128] hidden units, mapping the robot state feature and latent command to 12-dim robot target joint angles. Our low-level value network shares the same structure while taking a motion embedding as additional input, and output 1-dim value for RL training. Our learnable motion embedding is a 64-dim vector for each reference motion.

**High-level Policy $\pi_{high}$ and Value Network $V_{high}$:** Our high-level is formulated as a three-layer MLP with [256, 128] hidden units, mapping proprioception information and high-level task information to high-level latent command for low-level motion prior. Our high-level value network shares the same structure. High-level task information depends on specific task. Additional implementation details are provided in the supplementary materials

**Reward Coefficients:** In our experiment, we use $w_{func}^{ori} = w_{func}^{pos-xy} = 0.1875$, $w_{func}^{pos-z} = 1.5$, $w_{style}^{adv} = 1$, $w_{style}^{joint} = 0.5$.

**Other Rewards:** To smooth the robot trajectory, we also include energy penalty $r_{energy}$, and action smooth reward $r_{action}$. $r_{energy} = -1e - 3 * \sum_i |\tau_i \times \dot{q}_i|$ where $\tau_i$ is the the motor torques applied to the $i$th joint, and the $\dot{q}_i$ is the joint velocity for the $i$th joint. $r_{action} = -1e - 2 * \sum_i |a_i^t - a_i^{t-1}|$ where $a_i^t$ and $a_i^{t-1}$ are the action from policy for $i$th joint at current timestep and the previous timestep.

**Simulation Setup:** We utilize IsaacGym[68] to simulate 4096 robots in parallel and our simulation runs in 200 Hz. During motion prior training, for each robot, we uniformly sample a reference motion from the dataset for it to track.

## G  High Level Task Reward

Our high level jumping reward is adapted from CAJun [59] with the following terms.

$$r_{jump} = 2 * (2 - \|v_{robot} - 2\|)/2 + 5 * (\text{Base Height} - 0.6) * \max(\sum_{f \in feet} \hat{c}_f - 4, 0)$$

$$+ 3 * \sum_{f \in feet} \|1 + c_f - \hat{c}_f\|^2 + 1 * \sum_{f \in feet} \|c_f - \hat{c}_f\| * \min(h^f, 0.16)/0.16$$

Here our desired foot contact $\hat{c}_f$ at each step is a binary value generated by the task generator as in CAJun [59] with value 0 for no contact, and value 1 for contact, similar for the actual contact $c_f$, and $h^f$ is the foot height over the ground.

Our high level command following reward is defined as follows.

$$r_{following\ cmd} = 1.5 * \exp\left(\|v_{command} - v_{robot}\|^2 / 0.25\right) + 1.5 * \exp\left(\|\omega_{command} - \omega_{robot}\|^2 / 0.25\right)$$

$$- 2 \|\text{Base Height} - \text{Target Height}\|$$

Here the $v_{command}$ is the commanded target linear forward velocity, $v_{robot}$ is the current forward velocity of the robot. The $\omega_{command}$ is the commanded target angular velocity, $\omega_{robot}$ is the current angular velocity of the robot.

We also include energy penalty and action smooth reward as shown in the other rewards in Appendix F.

## H    Additional Low-level Skill Comparison

In addition to the low-level skill comparison in Figure 5, we provide another low-level skill comparison in Figure 9. Our controller learned to jump forward in the air with a natural gait, while baselines failed to leave the ground or failed to move forward

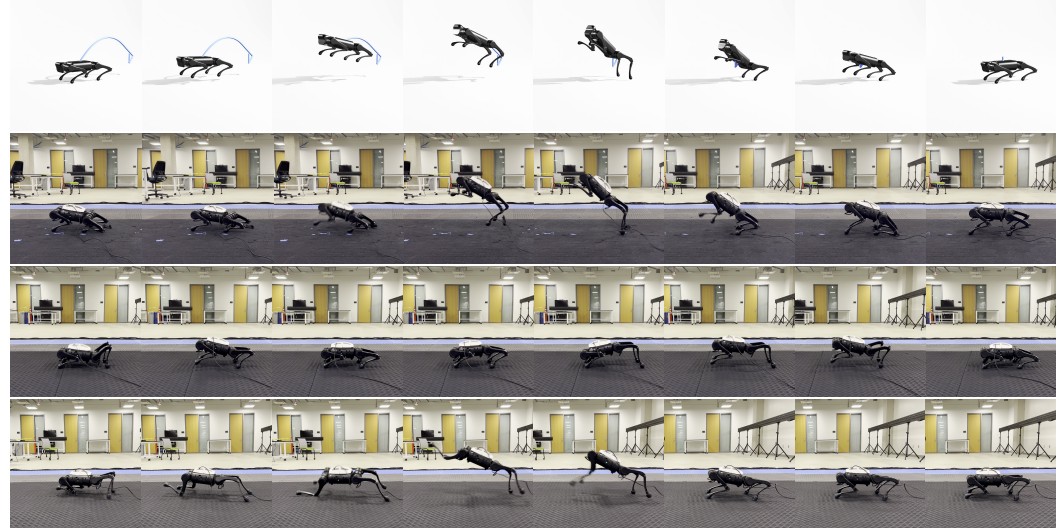

Figure 9: **Real World `Jump Forward` Trajectory Comparison:** Each row represents a single trajectory (From top to bottom: Reference Motion, VIM, GAIL, Motion Imitation). Trajectories are shown from right to left.

## I    High-level Policy Visualization

To better understand of the performance of our high-level policy, we provide *Following Command* trajectory in Figure 10. Though our low-level controller only learns to turn with specific angular velocity, our high-level could track different angular velocity in the real world.

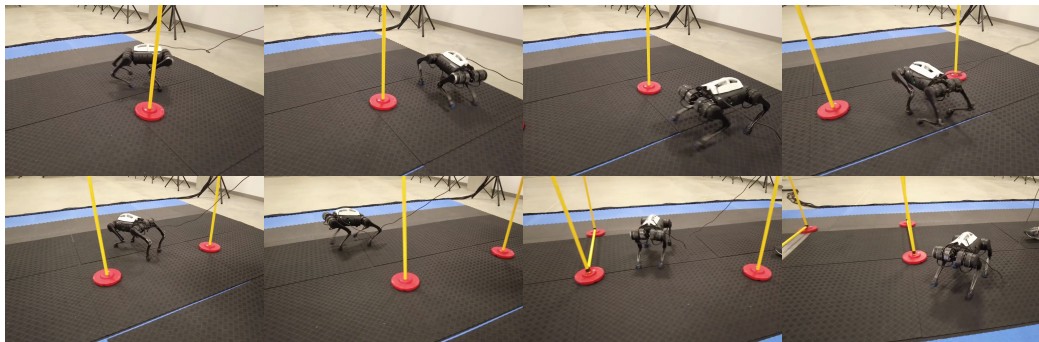

Figure 10: **Real World high-level Following Commands trajectory:** Our high-level Following Command policy can track wide-range linear and angular velocity commands even for velocities absent in the reference motion dataset, indicating high-level policy can manipulate the motion prior for High-level tasks. The trajectory is shown from left to right, from top to down.

## J    Detailed Observation Space

We provide more detailed observation space for our motion prior. Our Unitree A1 robot has 12 joints, corresponding to 12 Degrees of Freedom (DoF), and we use positional control for the 12 DOF ($KP = 40$ and $KD = 1.0$). Specifically, the proprioceptive state of the robot contains:

- **Joint Angle -** $\mathbb{R}^{12\times3}$ contains joint rotations for all joints (12D) for the past three control step.

- **Joint Velocity -** $\mathbb{R}^{12\times3}$ contains joint velocities for all joints (12D) for the past three control step.

- **Previous Action -** $\mathbb{R}^{12\times3}$ contains positional command for all joints (12D) for the past three control step.

- **Projected Gravity -** $\mathbb{R}^{3\times3}$ contains the projected gravity in the robot frame, representing the orientation of the robot for the past three control steps.

- **Foot position -** $\mathbb{R}^{3\times4\times3}$ contains the robot foot positions in the robot frame, 3 dim per foot per timestep for the past three control steps

Note that, our discriminators used for adversarial reward only use the joint angle transition for training and reward calculation.

We also provide additional high-level observation for the example high-level tasks we used. For *Following Command* task, we provide target linear velocity and target angular velocity as high-level observation. For *Jumping Forward* task, since the robot is tasked to jump forward in a fixed frequency, we provide normalized temporal phase as high-level observation.

## K   Domain Randomization

Here we provide our hyperparameters related to domain randomization for better sim2real transfer and shared by all methods.

| Parameter | Range |
|---|---|
| Added Mass for the base | [-1, 3] |
| Friction | [0.1, 1.3] |
| Restitution Range | [0, 1.0] |
| COM shift | [-0.05, 0.05] |
| Motor Strength ratio | [0.7, 1.1] |
| KP randomize ratio | [0.8, 1.5] |
| Kd randomize ratio | [0.5, 1.5] |
| Proprioception noise | [0, 0.01] |
| Action noise | [0, 0.05] |

## L   RL Training Details

Here we provide hyperparameters related to RL training and shared by all methods.

| Hyperparameter | Value |
|---|---|
| Max Episode Length | 500 |
| Non-linearity | ELU |
| Policy initialization | Standard Gaussian |
| # of samples per iteration | 4096 * 48 |
| Discount factor | .99 |
| Parallel Environment | 4096 |
| Optimization epochs | 5 |
| # of batches | 16 |
| Clip parameter | 0.1 |
| Policy network learning rate | 3e-4 |
| Value network learning rate | 3e-4 |
| Discriminator learning rate | 1e-5 |
| Entropy | 0.001 |
| Optimizer | Adam |
| $\beta$ for latent regularization | 1e-5 |

## M  Additional Results Regarding Stylization Tracking

Though adversarial training is generally unstable, we found it relatively stable during our training. We provide the training log of our discriminator and the average adversarial reward across epochs in Figure 11. Specifically, We applied the following techniques to stabilize the adversarial training.

- We clipped the gradient of the discriminator to have the maximum norm of $0.5$
- We applied gradient penalties during the training of the discriminator, following AMP [22]

We think our motion imitation reward also helped stabilize the adversarial training since our joint tracking and end-effector position tracking reward provide fine-grained instruction for policy training. We didn't observe specific latent skill space collapse during our training, we think this is for the following reasons:

- We provide motion embedding for the value function to distinguish different reference motions during training, as shown in Figure 3 in the manuscript. With motion embedding, the value function in our method could learn to distinguish different skills easily.
- Our tracking reward terms provided dense instruction for the controller to generate different behaviors, which further regularized the latent skill space during training.

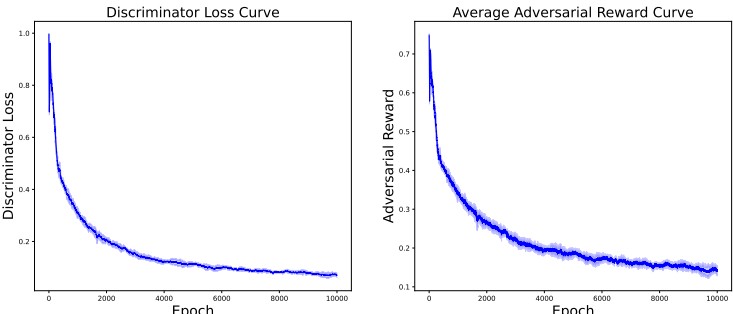

Figure 11: **Adversarial Training Log**

To study the training behavior in more detail, we visualized the episode return for the joint angle tracking term with the average adversarial reward in Figure 12. We found that in the first 1000 epoch, the episode return of joint tracking increased swiftly corresponding to the rapidly decreasing period of average adversarial reward. After 1000 epochs, the increasing rate of the episode return of joint tracking drops. We think this phenomenon corresponds to our claim in the manuscript that the model transits from learning the overall motion, where the episode return of joint tracking boost, towards learning the fine-grained behavior using the joint angle tracking reward, where the episode return of joint tracking grows slowly.

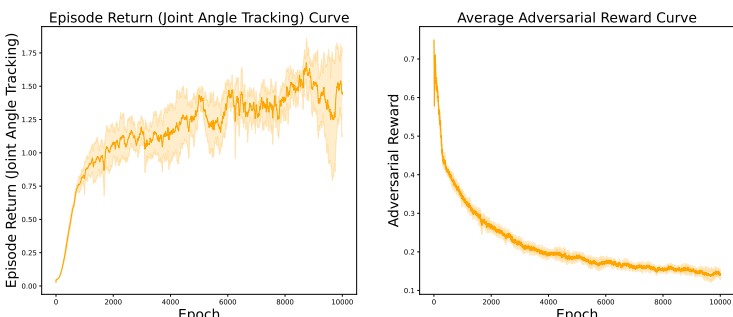

Figure 12: **Average Adversarial Reward with Episode Return (Joint Angle Tracking)**

