# OpenReview forum: "Generalized Animal Imitator: Agile Locomotion with Versatile Motion Prior"
_robot-learning.org/CoRL/2024/Conference — CoRL 2024_

### Official Review · Reviewer_wyry · 2024-06-28
**A reliable locomotion learning framework**

**Originality:** 3
**Technical Quality:** 4
**Clarity Of Presentation:** 4
**Potential Impact:** 3
**Recommendation:** 3
**Confidence:** 4

**Review:**

Overall, this work proposes a reliable method for learning a single policy that can exhibit multiple locomotion skills.

Strengths:
1. Extensive simulation experiments and quantitative comparisons validate the proposed method.
2. Solid hardware experiment results clearly show that the learned policy can be used to generate diverse motions on a hardware robot.

Weaknesses:
1. The method proposed in this work is not particularly novel; similar works such as [1][2] have been proposed. However, these works do not validate on real-world robots.
2. The clarity of the paper can be improved. For instance, in Table 1, some metrics are unclear. For example, what is meant by agility, and why can't ASE capture agile motion? I suggest that the authors illustrate these metrics and different methods in the supplementary materials.
3. The motivation is somewhat unclear. Why is it beneficial to learn a single controller to capture all motions instead of multiple sub-policies? In [3], different locomotion skills are separately learned, and the results demonstrate super agile movements.
4. There are missing references. [4] also learns a low-level controller and a latent space by imitating reference motions.


[1] Luo, et al. Universal Humanoid Motion Representations for Physics-Based Control.
[2] Luo, et al. Perpetual Humanoid Control for Real-time Simulated Avatars.
[3] Hoeller, et al. Anymal parkour: Learning agile navigation for quadrupedal robots.
[4] Li, et al. Planning in learned latent action spaces for generalizable legged locomotion.

**Quality Of The Limitations Section:**

3

**Questions For Rebuttal:**

1.Weakness No. 3 is my main concern. The authors should focus on justifying the benefit of learning a single policy versus multiple sub-policies.
2. I am also interested in how the proposed method compares to baseline methods in learning a single reference trajectory.

**Robotics Focus:**

4

**Summary Of Paper:**

This work addresses the challenge of learning a single reusable quadrupedal robot motion prior by imitating diverse reference motion trajectories. The proposed Versatile Instructable Motion (VIM) framework achieves this by learning to project reference trajectories into a latent space. By varying the latent vector, diverse motions can be generated through a learned low-level policy. The framework is trained using both a functionality reward and a style reward. Validation of the method was conducted in both simulation and hardware. The results demonstrate that the proposed method outperforms various baseline methods and successfully generates diverse motions on an A1 robot.

**Summary Of Recommendation:**

I am towards a weak acceptance due to the overall quality of the hardware valuations as well as the reliable of the proposed method.

---

### Official Review · Reviewer_M6Lk · 2024-07-06

**Originality:** 3
**Technical Quality:** 3
**Clarity Of Presentation:** 4
**Potential Impact:** 3
**Recommendation:** 3
**Confidence:** 4

**Review:**

Key components of the system:

1 A style reward that is a mix of AMP style reward and imitation-based reward, with a learning schedule to adjust the relative weighting between them.

2. A reference motion encoder that encodes the reference motion, and a high level policy to output the encoding to learn high level tasks.

Pro:
1. real-world demonstration of various quadrupedal skills.
2. A latent space that enables transitions between skills.

Cons:
1. I am surprised the baseline method cannot learn some of the skills demonstrated as many works have shown they can, e.g.,
(1)Adversarial motion priors make good substitutes for complex reward functions, IROS 2022. AMP reward is able to get the robot trotting and pacing.
(2) Lifelike Agility and Play on Quadrupedal Robots using Reinforcement Learning and Generative Pre-trained Models, using imitation-based reward can also do jumping.
I am especially concerned about the trotting being a failure in the baseline methods, as trotting is pretty easy.

2. There are some skills that are very interesting, like jumping while running and cantering. However, the generated motion is not as nice looking as the reference.

3. The description and experimentation on the high level policy part of the paper is limited.

**Quality Of The Limitations Section:**

3

**Questions For Rebuttal:**

1. "To the best of our knowledge, this is the first work that allows a robot to concurrently learn diverse agile locomotion skills using a single learning-based controller in the real world.". I think "Lifelike Agility and Play on Quadrupedal Robots using Reinforcement Learning and Generative Pre-trained Models" also learns diverse locomotion skills with a single policy, and has been on arxiv since August 2023.

2. Some concerns I listed in the review sections.

**Robotics Focus:**

4

**Summary Of Paper:**

This paper presents a system that can train a robot to imitate multiple reference motions as well as performing transitions between them.

**Summary Of Recommendation:**

I think this is a good attempt at generating diverse agile skills for quadruped robot. Although the potential of the system is fully explored, e.g, solving more intersting high level tasks, get nicer looking low level skills.

---

### Official Review · Reviewer_wE5L · 2024-07-28
**Core idea is Adversarial imitation + joint and root tracking. But has some major issues that need to be addressed**

**Originality:** 2
**Technical Quality:** 2
**Clarity Of Presentation:** 2
**Potential Impact:** 2
**Recommendation:** 3
**Confidence:** 4

**Review:**

The paper presents an approach to train a single policy to perform multiple skills on a quadruped robot. While the authors show some evidence of sim-to-real transfer and improvement over baselines, I have some concerns about the approach:

## Major concerns:

1) Transition between different skills: One of the advantages of learning a single policy for multiple tasks, is the ability to compose different skills by switching between them as required. It is unclear how the policy can learn this with the current framework. Is the policy exposed to different skill transitions during training? If yes, what happens to $z_t$ during such transitions? To showcase the ability to switch skills, the authors should present more experiments in addition to running-jumping. Also, the hardware results for just jumping is better than running to jumping (the robot does not land with its front feet). I feels like the policy is abruptly switching skills without learning anything about the optimal strategy to switch.

2) Position and joint angle tracking: Why do we want to track the orientation/xy position of the reference trajectory? Isn’t this unnecessarily increasing the complexity of the task? for example, if the reference has a trajectory where the robot jumps from (x=1,y=0) the learned policy can also only jump at (x=1,y=0). It is much more useful to have a policy that can jump at any position.

3) A fundamental problem with the way the tracking rewards seem to be structured is that they break the MDP assumption. Consider a robot is standing in the nominal pose (same state) at two different points in the reference trajectory $t_a$ and $t_b$. Lets say from $t_a$ point the robot moves its left leg forward and at $t_b$ it moves the right leg forward. According the the policy both these states are indistinguishable. But for the same action the policy will get a different reward based on what the current $t$ is. This violates the markovian property. Usually the tracking reward comes with a phase term to overcome this limitation.

4) Instead of using GAIL, using AMP with a single skill is a stronger baseline for the low-level skills. Also, in Table 4. does AMP fail to jump in (Jump Distance)? I believe that this task is simple enough for AMP to be able to solve and might be an implementation/hardware issue. In addition, there seems to be an asymmetry between left and right turning which the authors should explain. How many trials were done for the real world deployments?

## Minor remarks:
1. The paper has a quite a few typos and grammatical mistakes. I encourage the authors to proofread the paper thoroughly. I've listed some of the below:

line 50 - "We use three feedback"

line 85 - "contructing"

line 89 - "quadrupedsBBB"

line 204 - "Funtionality"

2. For HRL, the authors claim that using a trajectory generator is unfair for the purposes of comparison - But from my understanding the proposed method uses trajectories generated by optimization as reference. I'm not sure if this is justified.
3. In line 149-  "joint angle tracking reward [51, 17] provides sparse but stable instruction for the robot to mimic" Why is this reward sparse?
4. Lines 137-139 need citation or should be removed

**Quality Of The Limitations Section:**

2

**Questions For Rebuttal:**

Listed above.

**Robotics Focus:**

4

**Summary Of Paper:**

This paper proposes a technique to train single policy for a collection of skills on a legged robot. The approach consists of two types of reward: 1. functionality and 2) style reward. The functionality reward tracks the root orientation and position(z). The style reward has two components that 1) tracks the joint angle and 2) captures the style using N discriminators (1 for each skills) to capture the distribution of state transitions (similar to AMP). The authors claim that their method works better than vanilla PPO, AMP and HRL and provide experiments in sim and real to justify their claims

**Summary Of Recommendation:**

The paper seem to advocate for a mix of AMP like style reward and tracking rewards to train skills. But the current experiments and arguments presented in the paper do not make a convincing case.

---

### Author Rebuttal · Authors · 2024-08-13

Revised manuscript with modified part highlighted with red.

---

### Decision · Program_Chairs · 2024-09-04

**Decision:**

Accept

**Comment:**

The paper presents a novel algorithm for learning a single policy that can imitate a wide range of motor skills. While reviewers enjoy the results, there exist several concerns, such as (1) novelty against the baselines, (2) poor quality of the baseline approaches, and (3) missing technical details. We encourage the authors to discuss these issues in the rebuttal period.

During the rebuttal, the authors provided a lot of complementary experimental results against the baselines. It resolved a lot of concerns from the reviewers, and one updated his/her recommendation to WA. Based on these positive feedback, we recommend this paper to CoRL this year.